# Statistical Analysis of Polymer Nanocomposites for Mechanical Properties

**DOI:** 10.3390/molecules26144135

**Published:** 2021-07-07

**Authors:** Shankar A. Hallad, N. R. Banapurmath, T. M. Yunus Khan, M. A. Umarfarooq, Manzoore Elahi M. Soudagar, Anand M. Hunashyal, Sandeep V. Gujjar, Jayachandra S. Yaradoddi, Sharanabasava V. Ganachari, Ashraf Elfasakhany, Md Irfanul Haque Siddiqui, Masood Ashraf Ali

**Affiliations:** 1Center for Material Science, KLE Technological University, Hubballi 580031, India; nrbanapurmath@gmail.com (N.R.B.); amhunashyal@kletech.ac.in (A.M.H.); jayachandra@kletech.ac.in (J.S.Y.); sharanu14@gmail.com (S.V.G.); 2Department of Mechanical Engineering, KLE Technological University, Hubballi 580031, India; sandeep.gujjar@rediffmail.com; 3Mechanical Engineering Department, College of Engineering, King Khalid University, P.O. Box 394, Abha 61421, Saudi Arabia; yunus.tatagar@gmail.com; 4Department of Mechanical Engineering, SDM College of Engineering & Technology, Dharwad 580002, India; umarfarooq.ma@gmail.com; 5Department of Mechanical Engineering, Glocal University, Delhi-Yamunotri Marg, SH-57, Mirzapur Pole, Saharanpur 247121, India; me.soudagar@gmail.com; 6Department of Civil Engineering, KLE Technological University, Hubballi 580031, India; 7Mechanical Engineering Department, College of Engineering, Taif University, P.O. Box 11099, Taif 21944, Saudi Arabia; ashr12000@yahoo.com or; 8Department of Mechanical Engineering, College of Engineering, King Saud University, Riyadh 11451, Saudi Arabia; msiddiqui2.c@ksu.edu.sa; 9Department of Mechanical Engineering, College of Engineering, Prince Sattam Bin Abdulaziz University, Al Kharj 16273, Saudi Arabia; mas.ali@psau.edu.sa

**Keywords:** multi-walled carbon nanotube, epoxy nanocomposites, flexural test, scanning electron microscopy, energy-dispersive X-ray spectroscopy

## Abstract

Epoxy resins, due to their high stiffness, ease of processing, good heat, and chemical resistance obtained from cross-linked structures, have found applications in electronics, adhesives coatings, industrial tooling, and aeronautic and automotive industries. These resins are inherently brittle, which has limited their further application. The emphasis of this study is to improve the properties of the epoxy resin with a low-concentration (up to 0.4% by weight) addition of Multi-Walled Carbon Nanotubes (MWCNTs). Mechanical characterization of the modified composites was conducted to study the effect of MWCNTs infusion in the epoxy resin. Nanocomposites samples showed significantly higher tensile strength and fracture toughness compared to pure epoxy samples. The morphological studies of the modified composites were studied using Scanning Electron Microscopy (SEM).

## 1. Introduction

Nanoscience paves new avenues in the field of the scientific community as well as industry. In the case of civil engineering, a substantial interest can be observed in nano-sized materials, such as nano-silica powder, nanofibers, and nanotubes intended for structural applications [1]. Nanofibers and nanotubes are used to introduce nano-reinforcements in polymer matrices that enable a decrease in the required amount of steel reinforcement and reduce corrosion problems affecting engineering structures [2]. Diamond-like carbon material (DLC), such as carbon nanotubes (CNTs), possesses outstanding mechanical properties, exceptional stiffness, and high strength-weight ratios and toughness; these outstanding properties result in the potential of CNTs as ultimate reinforcing materials for the development of nanocomposites [3]. One of the major problems with polymer-based composite is its intrinsically brittle type of failure, which is due to their minuscule tensile strength compared to their compressive strength and high fracture toughness [4]. In appreciation of this flaw and to enhance polymer-based composites, extensive experimentation with filler reinforced composites has been carried out and encouraged [5]. The fillers can affect properties such as toughness, impact resistance, fatigue endurance, and the onset of crack initiation, in addition to the strength of the composite [6]. Today’s world demands multi-functionality, reduced cost, increased durability, and much more for composite materials [7]. The promise of being lighter, thinner, and stronger, with durability, makes CNT reinforced composites one of the most attractive types of filler reinforced composite [8]. Garg et al. [9] studied the effect of functionalized nanoparticles (NPs) in influencing the mechanical properties under study. The resin transfer method was employed to fabricate polymer nanocomposites consisting of Glass fiber/epoxy/MWCNTs and reported an increase in flexural strength by 155% [10]. A nano-thin film composed of MWCNTs showed improved strength of metallopolymer systems [11]. An attempt was made to investigate the mechanical/electrical properties of MWCNTs/epoxy composites. The optimum mechanical properties were observed at 0.1 wt.% and 0.25 wt.% for tensile strength and flexural modulus of modified composites. An increase in tensile modulus and decrease in strain to failure was observed with the increase in CNT content. The maximum performance in flexural strength was noticed at 0.05%, and the electric response of the modified composites was noted to a threshold value of 0.5% multi-walled carbon nanotube addition. A fine distribution of MWCNTs remains essential to improving mechanical properties under study. Poor distribution of nanoparticles fails the specimen because of the agglomeration of the CNTs. The agglomeration has less influence on the electrical properties of modified composites [12]. The mechanical/electrical properties of MWCNT-modified composites were investigated. Based on the results obtained, it was noticed that tensile strength and flexural modulus were optimum at MWCNT contents of 0.1% and 0.25% (mass fraction). With the increase in MWCNT dosage increase in tensile modulus and a decrease in the strain to failure, the ratio was noticed. This phase shift from plastic to the brittle mode of failure was noticed because of the upsurge of the MWCNT ratio. The optimum flexural strength of the MWCNT modified sample was found to be optimum at 0.05 wt.% of the base matrix. Compared to other reinforcement proportions, the electrical percolation threshold of the modified composites was found to occur at 0.5 wt.% of MWCNT addition. The homogeneous dispersion of the reinforcement filler is an important parameter in improving the mechanical properties of the modified composites. Homogeneous dispersion of nanoparticles is very much essential for enhancing the strength of the polymer composites. Agglomeration of infused reinforcement leads to the early failure of modified composites. The agglomeration has the least influence on the electrical properties of the modified composites [13,14]. The effect of incorporating the MWCNTs bucky paper (CNTBP) and MWCNTs bucky paper/Epoxy (CNTBPE) on flexural strength of three-ply glass fiber/epoxy (3GF) was investigated. CNTBPE samples exhibited an improvement in flexural strength by 50% and 30% compared to 3GF and four-ply glass fiber composite, respectively. An increase in the specific modulus and strength by 30% and 70% compared to 3GF was observed [15]. The influence of varying MWCNTs (0.15, 0.25, 0.50, and 0.75 wt.%) on mechanical and thermal properties of multi-scale Carbon Fiber-Reinforced Plastic (CFRP) were investigated. Mechanical tests were conducted to study the influence of NPs on mechanical properties (tensile strength, modulii, and flexural strength) of CFRP. Thermal tests such as differential scanning calorimetry, thermo-gravimetric analysis, and dilatometry were conducted to investigate the effect of NP on glass transition temperature, the coefficient of thermal expansion (CTE), and the thermal stability of CFRP. SEM and the optical microscope were utilized to observe the dispersion of MWCNT in the holding matrix. The broken particles of the specimen were utilized for SEM observation. An improvement in the tensile, flexural strength, and Young′s modulus by 60%, 54%, and 26%, respectively, was observed with the addition of 0.25 wt.% of MWCNT in the CFRP composite. The optimum performance of the developed composite could be due to the NPs acting as nano-stitches holding the matrix together, thereby inhibiting the development of microcrack propagation during failure. The CTE of CFRP was reduced due to the addition of MWCNTs; the composites were thermally stable up to 350 °C. From the spectrographic observation, MWCNTs were homogeneously dispersed in the epoxy matrix until a certain dosage beyond which agglomeration of the fillers was observed [16].

Gantayat et al. [17] developed the nanocomposites by infusing functionalized MWCNTs (F-MWCNTs) in varying contents of 0.4, 0.6, and 1 wt.% into the epoxy. Nanocomposites were characterized by X-ray diffraction and Fourier Transform Infrared Spectroscopy (FTIR). The field emission scanning microscope was used to study the morphology of developed composites. Nanocomposites loaded with 0.6 wt.% functionalized MWCNTs exhibited higher tensile strength and Young′s modulus compared to other composites. This enhancement can be attributed to the good dispersion of the NPs in epoxy. An intimate bonding between the filler and the matrix was also responsible for the enhancement of the mechanical properties. Swam et al. [18] studied the effect of F-MWCNTs against MWCNTs into the epoxy matrix. The F-MWCNTs composites were prepared by varying compositions of MWCNTs of different weight percentages (0.4, 0.6, and 1 wt.%) into the epoxy resin. Thermal mechanical and electrical properties were improved because of the incorporation of MWCNTs. Zabini et al. [19] studied the effect of amine-functionalized and pure CNTs infusion into the epoxy matrix to study the effect of F-CNT influence on thermal, mechanical, and morphological properties. Based on the results, it can be obtained that amine-functionalized CNTs showed improved glass transition temperature, tensile properties, and thermal stability of modified composites. Jin et al. [20] prepared MWCNTs based Carbon/Carbon composites by employing the chemical vapor infiltration technique. The electrical conductivity and compressive strength reached 191 S/cm and 148.6 MPa, respectively. This research aims to improve the properties of Epoxy resin with the direct addition of MWCNTs in low concentrations. Epoxy samples blended with MWCNTs are prepared, and the effect of NPs on epoxy properties is investigated with fracture and mechanical characterization.

## 2. Materials and Methods

The material used in the present investigation is Epoxy resin and MWCNTs. The epoxy resin used to carry out this work is Bisphenol-A (L-12) and associated hardener L-aliphatic amine (K-6) (Supplied by Atul Industries Ltd., Gujarat, India). MWCNTs were manufactured by the Chemical Vapor Deposition method [21], which resulted in tubes with a diameter in the range of 10–30 nm in diameter. The length is 1–2 µm with carbon purity content greater than 95% in volume and surface area greater than 350 m^2^/g (procured from Sigma-Aldrich, St. Louis, MO, USA).

### 2.1. Preparation of Nanocomposites

Pure epoxy nanocomposites loaded with MWCNTs in proportions of 0.1–0.4 wt.% in the incremental steps of 0.1 to matrix weight were synthesized. The steps followed during the preparation of these nanocomposites are shown in Figure 1.

First, MWCNTs, 250 gm of epoxy resin, 25 gm of curing agent, along with 0.5 mL of degassed solvent are mixed to form a mixture. The mixture is sonicated using a bath for 120 min to minimize the agglomeration of MWCNTs and to ensure uniform dispersal of NPs in the matrix. As the epoxy resin passed between three rollers, it was subjected to a shear force that untangled the NPs and dispensed the dispersed NPs into the epoxy resin. The curing agent is added to the mixture and stirred for 5 min at a speed of 350 rpm by a mechanical agitator. The mixture is then placed in a vacuum chamber coupled in an oven for 20 min and then exposed to atmospheric pressure for 10 min. This process is employed to reduce the viscosity of the epoxy resin that relieves any trapped bubbles in the agitated mixture, thereby enhancing the fluidity. A pressure of 150 kN/m^2^ is applied for the curing specimen to obtain a flat plate. The mixture is then cooled for 2 h. Finally, MWCNTs reinforced epoxy is poured into the mold and cured at room temperature [22]. Nanocomposites are coded based on their composition and are enlisted in Table 1.

### 2.2. Tensile Testing of Nanocomposites

Tensile testing of the PE and nanocomposites were carried out as per ASTM D 3039 standard using a dog bone-shaped specimen shown in Figure 2. These tests were carried out using Tinius Olsen UTM (10 kN capacity) with fixtures movement at 1 mm/min. The stress–strain plots obtained from tensile tests were used to determine Young’s Modulus (MPa), tensile strength (MPa), and % strain failure. Five specimens from apiece nanocomposite variation and PE were tested, then the results are averaged. The dimension of the tensile test specimen used was 250 mm × 25 mm × 2.5 mm.

## 3. Results and Discussions

The composite beams were assessed for strength-deflection criteria. The results were weighed against the values of the pristine specimens. The outcomes are presented in Table 2.

The load curves are shown in Figure 3. From Figure 3, it follows that the ductility is in direct proportion to the MWCNT strength fillers in the matrix linearly up to 0.3 wt.% of fillers, beyond which it decreased drastically.

The reasons could be due to the high aspect ratios of MWCNTs which form the significant factor in ductility enhancement. This can also be attributed to the flexibility of the MWCNTs and the rehybridization capacity of the carbon atoms in the degree of sp^2^-sp^3^. For 0.3 wt.% of MWCNTs infusion in the resin matrix, tensile strength attained was 63.0 N/mm^2^ against neat epoxy of 24.8 N/mm^2^, an upsurge of 154%. The dosage of MWCNTs beyond 0.3 wt.% up to 0.4 wt.% tensile strength decreased, and this could be due to the sword in sheath mechanism of failure (telescopic extension nature of multi-walled carbon nanotubes that relieves stress deposition till stress in proportion to strain) means the stress was primarily concentrated on the outside walls of the MWCNTs when compared to the inner walls before rupture [22].

### 3.1. Machine Learning Analysis on Tensile Test

The experiment fits the best curve for polynomial regression when the degree of freedom was taken to 5, as shown in Figure 3. The details of the nodal values considered are mentioned in Table 3. This curve best fits the experimentation curve with an R-square (COD) value of 1.

The tensile stress vs. strain plots is depicted in Figure 4, which shows the outcome of MWCNT′s addition on tensile properties of composites [23].

The tensile strength of all samples increased with the addition of the MWCNTs, and the changes in strength were almost similar [24]. From the stress-strain curve depicted in Figure 4, the area under the curvature for specimen A3 was higher when compared to other compositions. From this, it can be concluded that 0.3 wt.% was the best composition for tensile strength when compared to the other reinforcement compositions.

### 3.2. Machine Learning Analysis on Stress vs. Strain Plots of Tensile Test

The experiment fits the best curve for linear regression, as shown in Table 4. Based on the linear fit analysis, it can be shown that the experimentation has been carried out with simple linear regression analysis to study the interaction between the dependent variable stress and the independent variable strain. Figure 4 shows the parameters concerning regression analysis. The residual difference between the experimental and theoretical values remains at 0.96721 with a standard deviation of 0.00232.

### 3.3. Flexural Testing of Nanocomposites

Flexural properties of PE and nanocomposites were determined using the three-point bending method as per ASTM D 790 standard. The specimen used for conducting the test are as shown in Figure 5.

The tests were conducted on Tinius Olsen UTM (10 kN capacity) using a crosshead speed of 1 mm/min with a width of 30 mm. The flexural modulus and strength were determined from these tests. Five specimens from each variation were tested, and the results were averaged. A support gap to depth ratio of 16:1 is maintained. The loading is made till the rupture of the external surface of the sample or till a strain reaches a maximum value of 5%, whichever is earlier. A strain extent of 0.01 mm/min is maintained to apply a gradual load on the trial sample. The machine was calibrated so that the fault in the capacity measurement does not vary more than ±1% of the max load that is to be quantified.

### 3.4. Flexural Tests

The load-deflection plots obtained from the three-point bending tests of PE and MWCNTs loaded epoxy composite are shown in Figure 6. The load-displacement plots are shifted horizontally for the simplicity of presentation. The results of flexural tests are tabulated in Table 5. From Table 5, observations flexural strength (FS) of epoxy loaded MWCNT showed inclining trends with an increase in the concentration of filler up to 0.3 wt.% compared to PE laminates, whereas EM4 laminates exhibit lower strength wrt base epoxy laminates. The FS of PE laminates is 433.33 MPa, and FS increased by 150% to 1094 MPa with the addition of 0.3 wt.% of MWCNTs to epoxy. The cause for this observed drift could be endorsed by the good dispersion of the MWCNTs in facilitating the transfer of applied load to the nanofillers by the polymer matrix [25]. This can also be attributed to the good adhesive force between polymer matrix chains at an optimum proportion of reinforcement [26]. Beyond 0.3 wt.% of a filler, falling trends in strength facets were noticed, which may be due to the agglomeration of MWCNTs in the holding matrix [27].

### 3.5. Morphological Characteristics

Morphological studies, such as SEM and EDX, are conducted to study the effective interfacial adhesion between the holding matrix and the strength fillers [28]. Scanning electron microscopy and EDX were observed for sample (EM1-4) for surface binding between the fillers and the matrix.

### 3.6. Scanning Electron Microscopy

The rupture ace of epoxy composites altered by MWCNTs was characterized by SEM to study the dispersion of the MWCNTs in the holding matrix. Figure 7, Figure 8, Figure 9 and Figure 10 shows the SEM images of the epoxy/MWCNTs modified composites. The area encircled with the blue lines indicates a good interfacial adhesion between the holding matrix and the fillers. The shiny region marked the blue lines depict the reflection of the light source because of the presence of Diamond-Like Carbon (DLC) entities, such as MWCNTs nanoparticles. The breakage of the polymer chains at a higher level of reinforcement can be seen in Figure 10, which also indicates the poor distribution of MWCNTs and the lack of interfacial adhesion between the polymer chains [29,30,31,32]. The reinforcement at 0.3 wt.% of the epoxy/MWCNTs, as shown in Figure 9, indicates good dispersion and good interfacial linkage in-between the fillers-matrix.

### 3.7. Energy Dispersive X-ray Analysis

EDX analyses were conducted on the selected samples to verify the metallic constituents of CNTs that influence mechanical properties under study. The metallic traces present in the fillers are shown in Figure 11. The presence of elements such as minute traces of oxygen and carbon are shown in Table 6 and are identified as the main elements within CNTs composition. Since the amount of sodium, niobium, and chlorine present in the CNTs is very small, depicted by the red line in Figure 11 (less than 1%), the effect of these elements affecting the mechanical properties understudy is neglected.

## 4. Conclusions

From the above experiment, we can conclude that the highly flexible behavior of CNTs has made them an ideal reinforcing phase. In the case of a tensile test, comparing the results of plain polymer with composites reinforced with epoxy/MWCNTs, that is, in the case of specimen A3 (0.3 wt.%), it is observed that an increase in tensile strength by 61% as compared to the unfilled epoxy resin. This could be due to the aspect ratio of MWCNTs, flexibility, and rehybridization capacity of the planar graphene sheet that makes up MWCNTs. Composites with a higher percentage of CNTs showed declining trends of tensile strength, which can be attributed to the non-homogeneous dispersal of MWCNTs. In the case of flexural strength, comparing the results of plain polymer with composites reinforced with epoxy/MWCNTs, that is, in the case of specimen A4, it is observed that an increase in flexural strength by 60% as compared to the PE specimen is observed. This is due to the hindrance of the crack growth at the micro-level by bridging the crack at the nano-level itself. The reinforcement beyond A3 (0.3 wt.%), that is, A4, has shown deteriorating trends in the strength features. This may be because of nanotube waviness with the increase in filler content. Composites with a higher % of CNTs showed declining trends of tensile and flexural strength. SEM analysis shows the dispersion of MWCNTs in the polymer matrix, and EDX shows the presence of other elements in the sample, which will affect the strength of the composites.

## Figures and Tables

**Figure 1 molecules-26-04135-f001:**
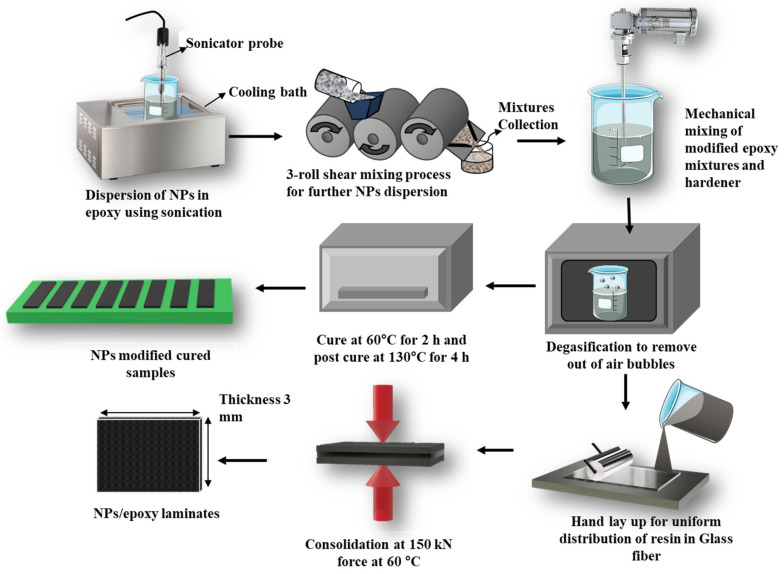
Epoxy nanocomposite fabrication by the hand casting process.

**Figure 2 molecules-26-04135-f002:**
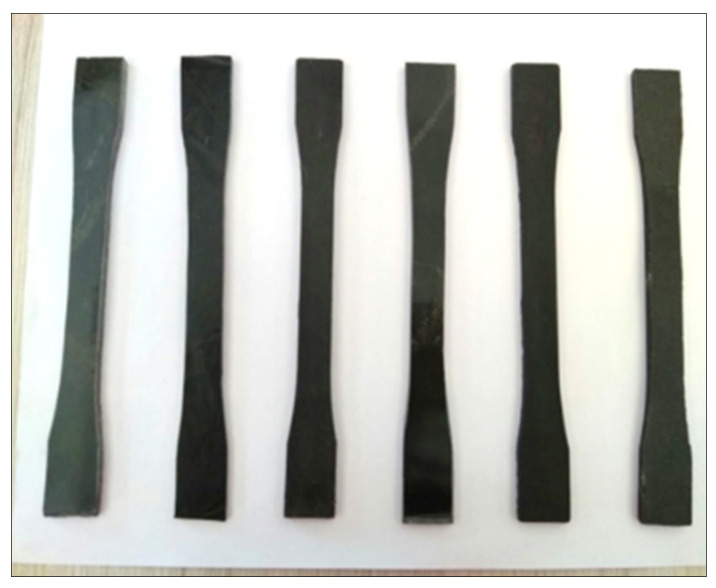
The tensile test specimen.

**Figure 3 molecules-26-04135-f003:**
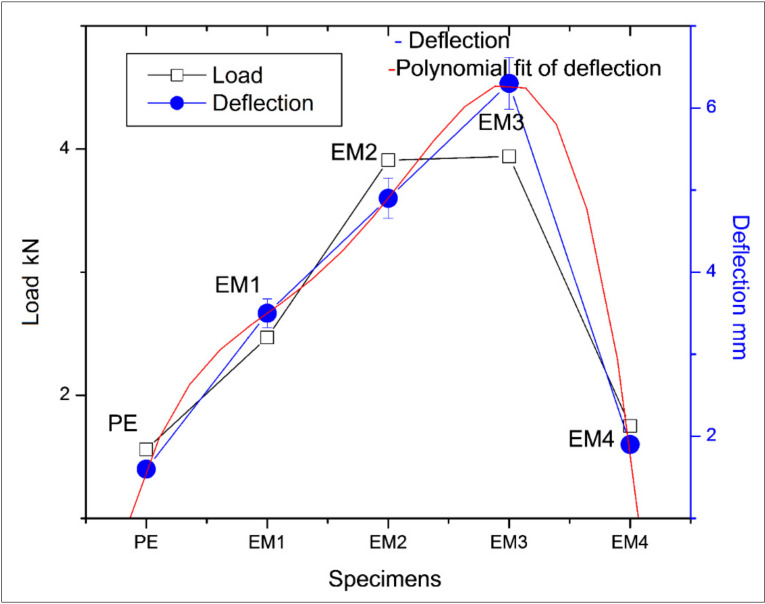
Tensile load and deflection of tensile testing.

**Figure 4 molecules-26-04135-f004:**
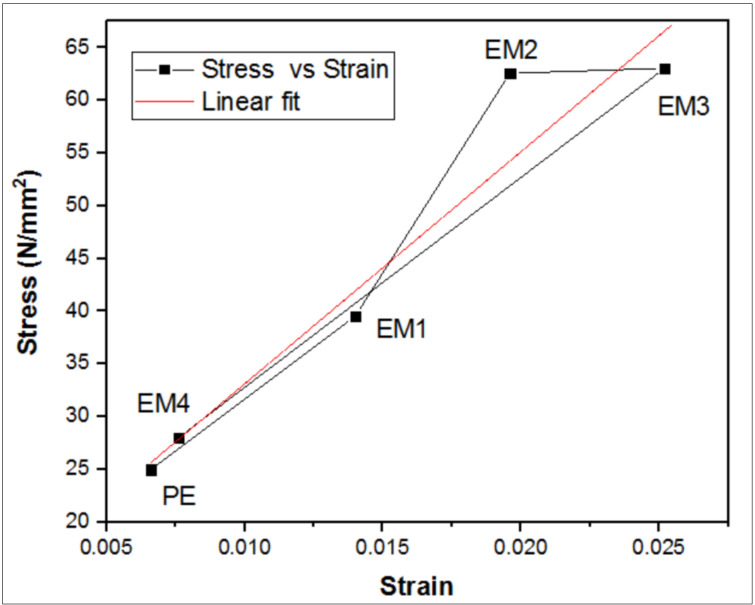
The stress vs. strain curve of tensile testing.

**Figure 5 molecules-26-04135-f005:**
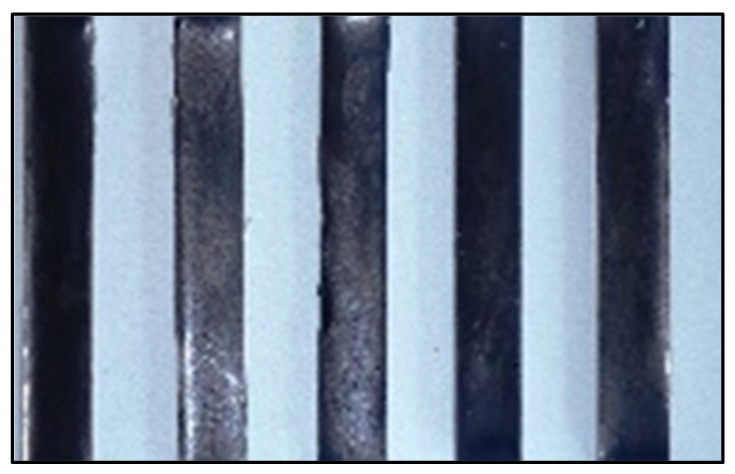
The specimen utilized for flexural testing.

**Figure 6 molecules-26-04135-f006:**
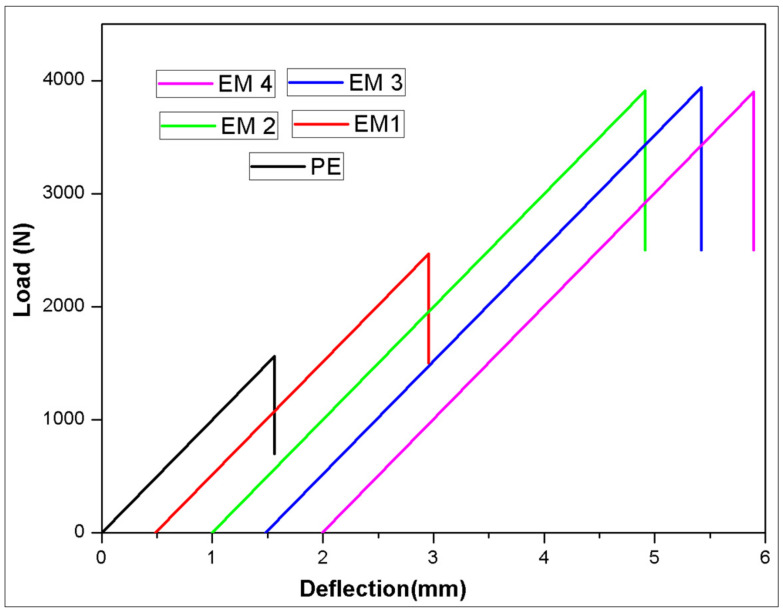
Load vs. deflection of flexural tests.

**Figure 7 molecules-26-04135-f007:**
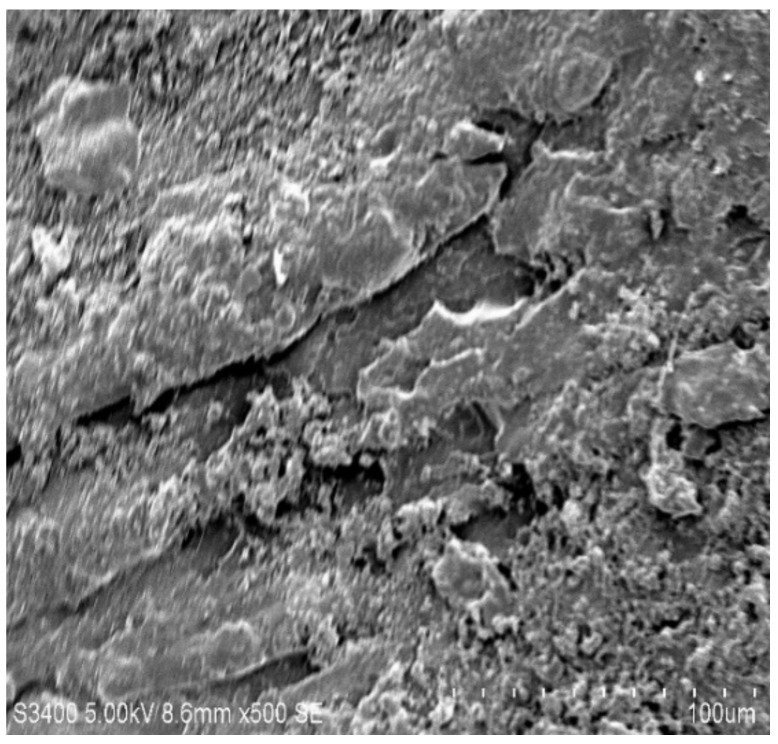
SEM of 0.1 wt.% MWCNT.

**Figure 8 molecules-26-04135-f008:**
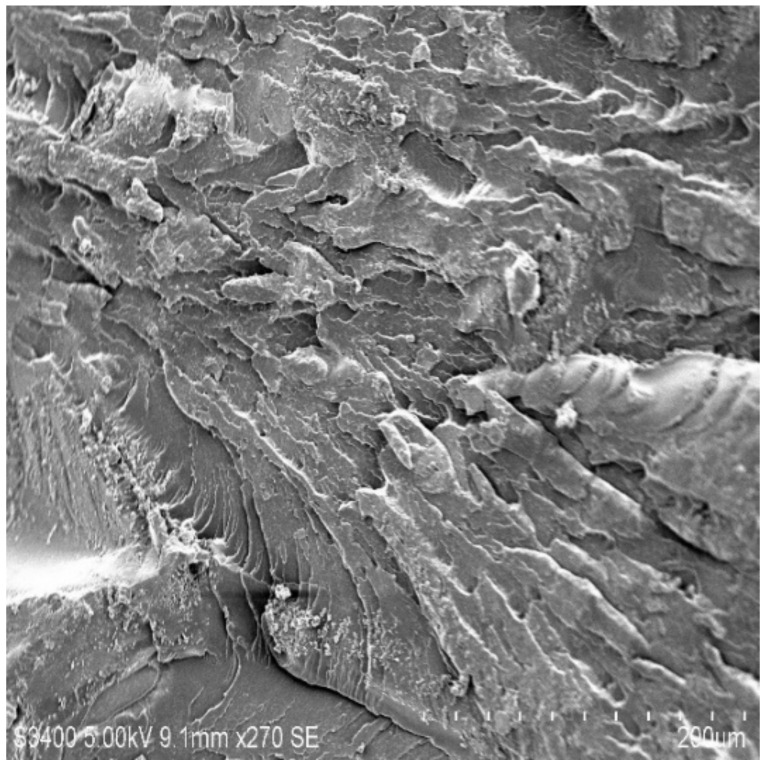
SEM of 0.2 wt.% MWCNT.

**Figure 9 molecules-26-04135-f009:**
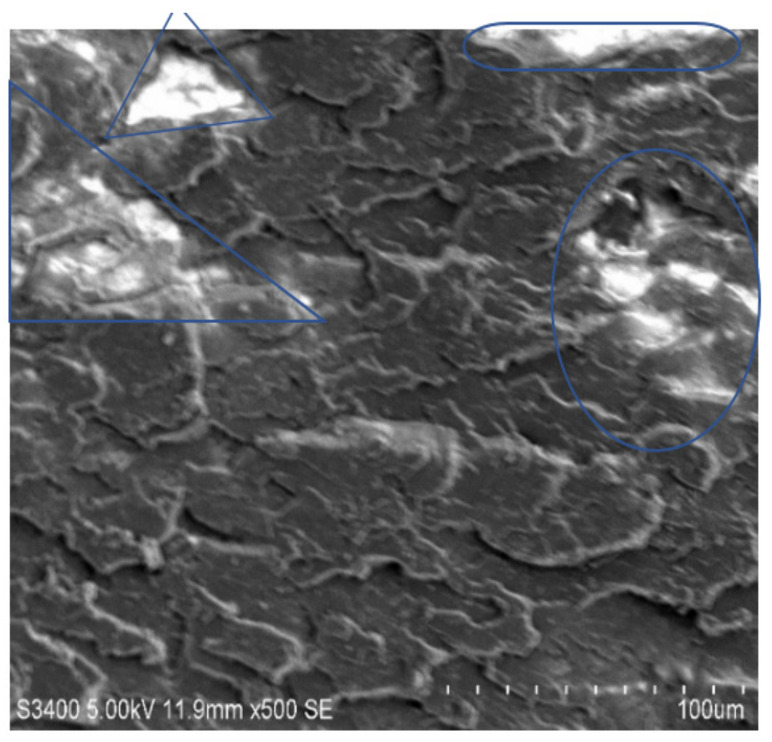
SEM of 0.3 wt.% MWCNT.

**Figure 10 molecules-26-04135-f010:**
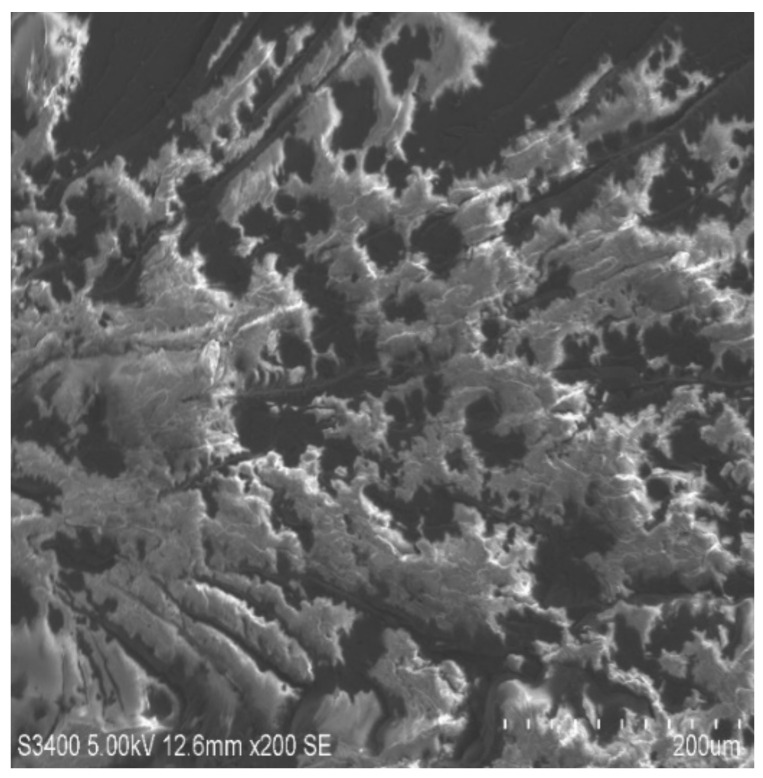
SEM of 0.4 wt.% MWCNT.

**Figure 11 molecules-26-04135-f011:**
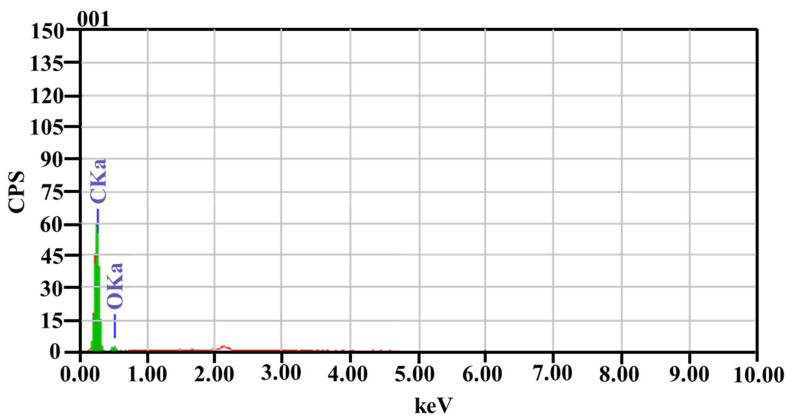
EDX of specimen A3.

**Table 1 molecules-26-04135-t001:** The coding of the test specimens.

Sl. No.	Specimen Composition	Specimen Coded
1	Plain Epoxy	PE
2	Epoxy + 0.1 wt.% (* w.r.t. Epoxy) of MWCNTs	EM1
3	Epoxy + 0.2 wt.% (w.r.t. Epoxy) of MWCNTs	EM2
4	Epoxy + 0.3 wt.% (w.r.t. Epoxy) of MWCNTs	EM3
5	Epoxy + 0.4 wt.% (w.r.t. Epoxy) of MWCNTs	EM4

* w.r.t.—with respect to the weight percentage of the epoxy resin.

**Table 2 molecules-26-04135-t002:** The results of the tensile tests.

Sl. No.	Composition	Ultimate Load (kN)	Ultimate Strength (N/mm^2^)	Deflection (mm)	Stress (N/mm^2^)	Strain	Young’s Modulus (N/mm^2^)
1	PE	1.56	24.81	1.6	24.96	0.0066	3782
2	EM1	2.47	42.24	3.5	39.52	0.0140	2823
3	EM2	3.91	62.56	4.9	62.56	0.0196	3345
4	EM3	3.94	63.04	6.3	63.04	0.0252	3680
5	EM4	1.75	28.00	1.9	28.00	0.0076	3684

**Table 3 molecules-26-04135-t003:** Machine learning analysis on the stress vs. strain plots of MWCNT modified specimens.

R-Square (COD)	SD	N	*p*
1	0	5	<0.0001

**Table 4 molecules-26-04135-t004:** Machine learning analysis on the stress vs. strain plots of MWCNT modified specimens.

R	SD	N	*p*
0.96721	0.00232	5	0.00709

**Table 5 molecules-26-04135-t005:** The results of three-point bend tests.

Sl. No	Specimen	Ultimate Load, N	Maximum Deflection in mm	Flexural Strength, MPa
1	PE	1560	1.56	433
2	EM1	2470	2.47	686
3	EM2	3910	3.91	1086
4	EM3	3940	3.94	1094
5	EM4	3900	3.90	1083

**Table 6 molecules-26-04135-t006:** Different elements present in epoxy/CNT.

Element	(keV)	Mass%	Error%	At%	% Composition
C	0.277	87.01	0.19	89.92	95.4910
O	0.525	12.99	2.60	10.08	4.5091
Total		100		100

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
