# Peer review of "Statistical Analysis of Polymer Nanocomposites for Mechanical Properties"

_molecules, 2021, doi:10.3390/molecules26144135_

Round 1

Reviewer 1 Report

The experiment design is reasonable. The conclusion is reliable. Further proposed changes include:

1) Delete the unnecessary diagrams, such as specimen sizes, and so on. Just describe it in words is enough.

2)The data of mechanical properties need to have error bar.

Author Response

The experiment design is reasonable. The conclusion is reliable. Further proposed changes include:

The authors take an opportunity to thank the reviewer for the generous words about our manuscript. In the following sections, you will find our responses to each of your comments and suggestions. We found the review comments very useful as we approached our revision, and the authors are grateful for the time and energy you expended on our behalf.

Comment 1: Delete the unnecessary diagrams, such as specimen sizes, and so on. Just describe it in words is enough.

Response: Thank you very much. The diagrams have been deleted as per your recommendations

Comment 2: The data of mechanical properties need to have error bar.

Response: Thank you very much for the recommendation, we have added the error bars as per your suggestion.

We found your comments extremely helpful in enhancing the quality of the paper. The authors are grateful for the time and energy you expended on our behalf.

Reviewer 2 Report

See attached file.

Author Response

Reviewer #2

The authors take an opportunity to thank the reviewer for the generous words about our manuscript. In the following sections, you will find our responses to each of your comments and suggestions. We found the review comments very useful as we approached our revision, and the authors are grateful for the time and energy you expended on our behalf. AR in the table implies “Author’s response”. The corrections made have been highlighted in the manuscript.

Comment 1:

“34”    Possesses

AR 1:

Has been corrected in the manuscript.

Comment 2:

“65”    Sentence is not clear, some words are missing.

AR 2:

Sentences have been modified and added in the manuscript.

Comment 3:

“64 & 68” Optimum for flexural strength at 0.1 and 0.25% according to line 64, but 0.05%

according to line 68. Please clarify.

AR 3:

Compared to other reinforcement proportions the electrical percolation threshold of the modified composites was found to occur at 0.5wt.% of MWCNT addition. The homogeneous dispersion of the reinforcement filler is an important parameter in improving the mechanical properties of the modified composites.

Comment 4:

“74” Sentence not clear, some words are missing.

AR 4

Sentences have been modified and corrected in the manuscript.

Comment 5:

“75” Please explain better, not clear.

AR 5:

Sentences have been modified and added in the manuscript.

Comment 6:

“75- 83”This whole paragraph is confusing; please reformulate and explain better.

AR 6:

Paragraph has been reformulated and added in the manuscript.

Comment 7:

“83-85” Sentence is not clear, some words are missing.

AR 7:

Sentences has been rephrased to give specific meaning.

Comment 8:

“81” MPa/gcm3: what is this; please use SI units.

AR 8:

Has been removed and rephrased in comparison with 3GF.

Comment 9:

“86” Tensile and flexural strength and moduli

AR 9:

Tensile strength, flexural strength and moduli has been added.

Comment 10:

“92”    CFPR: please explain abbreviation

AR 10:

Abbreviation has been added

Comment 11:

“nano stitches”: please explain better

AR 11:

Has been explained in manuscript.

Comment 12:

“94”    CTE: please explain abbreviation

AR 12:

Abbreviation has been added.

Comment 13:

“96”                3,500°C

AR 13:

3,500°C has been corrected to 350°C

Comment 14:

“99- 101” Sentence is not clear, please reformulate

AR 14:

Sentences have been reformulated.

Comment 15:

“103”  F-MWCNT: please explain abbreviation

AR 15:

Functionalized MWCNTs has been abbreviated as F-MWCNT

Comment 16:

“116”  “surface modified CNTs:  which modification was used?

AR 16:

The method  of composite fabrication has been added in the manuscript.

Comment 17:

“116-120”   What especially is the new approach in this work compared to what is already described in the literature? What is the assumed hypothesis of the work?

AR 17:

This work determines the optimum reinforcement of nanoparticles required to enhance the strength of nanocomposites. The reinforcement also increases the ductility of the plastic which is very much deserved for polymers in case of structural applications.

Comment 18:

“123”  L-12; K-6: give more (especially chemical) information.

AR 18:

Chemical name has been added in the manuscript.

Comment 19:

“124/125”       Add a reference to this method.

AR 19:

Reference has been added in manuscript

Comment 20:

127      m²/g

AR 20:

m2/g  has been changed to m2/g.

Comment 21:

Figure 1           Instead of “150 kN force” indicate the specific pressure. The force as such does not tell anything

AR 21:

Sentence inserted in section 2.1

Comment 22:

134      “degassed” instead of “degassing”

AR 22:

Has been corrected in Manuscript

Comment 23:

134      0.5 ml as sole information is useless; what was the amount of resin?

AR 23:

The amount of resin used has been added in manuscript.

Comment 24:

“136/137”       “shearing of the mixture”: what is the sense of this step?

AR 24:

As the epoxy resin is passed between three rollers, it is subjected to shear force which untangled the MWCNTs and dispense the dispersed MWCNTs into the epoxy resin.

Comment 25:

Table 1            “w.r.t.”: please explain

AR 25:

 w.r.t – with respect to the weight percentage of the epoxy resin, has been added in the manuscript.

Comment 26:

Figure 4           What means “C” in the graph?

                        Ad bars for standard deviation.

                        y-axis should be stress, not load.

                        What means “Data1_C”?

AR 26:

C – removed from the graph and error bars inserted

Comment 27:

172      63.0

AR 27:

Corrected

Comment 28:

173      24.8

AR 28:

Corrected

Comment 29:

175      “sword in sheath mechanism of failure: what does this mean? Explain better.

AR 29:

telescopic extension nature of multi-walled carbon nanotubes that relieves stress deposition till stress in proportion to strain.

Comment 30:

178      Is a polynomial regression really justified? You have a lot of turning points in such a curve. How can you explain them from experimental   standpoint?

AR 30:

This experiment best fits the polynomial regression anlaysis with R –square value equivalent to 1

Comment 30:

190      What does the “area under the curvature” mean? Why can you take it as the criterion to   judge that A3 is the best version?

AR30:

The load, strength and deflection are of high magnitude as compared to  other specimens as shown in Table.5

Comment 31:

198-200           Sentence is not clear, please reformulate.

AR 31:

Sentence reformed

Comment 32:

Figure 5, Where do you show this “variation of the parameters”? Please explain better.

AR 32:

Sentence modified as per the suggestion

Comment 33:

Figure 3, The samples look different. Please explain. and 6 

AR 33:

These samples are meant for flexural test so differ when compared to tensile specimens.

Comment 34:

212 “depth”: do you mean thickness?

AR 34:

Depth means thickness and depth has been replaced by thickness.

Comment 35:

Table 2 and Figure 4, belong to chapter 3, not to chapter 2

AR 35:

Modified as per suggestion

Comment 36:

221      Figure 7, not 5

AR 36:

Has been corrected

Comment 37:

225, “wrt”?

AR 37:

with respect to

Comment 38:

225, Strength of EM4 is still higher than for PE. Please clarify

AR 38:

The load, strength and deflection are of less magnitude as compared to EM3 as shown in Table.5

Comment 39:

Table 5, No digit after the comma for flexural strength

226 Add standard deviations

AR 39:

The mean of the five readings inculcated in data as the error bar coincides with the graph line

Comment 40:

226      1094.33 or 1094.44. But anyhow the correct result is 1094

AR 40:

Corrected as per suggestion

Comment 41:

226      FS: do you mean flexural strength?

AR 41:

Flexural strength (FS) abbreviated in section 3.1 line number 210

Comment 42:

230      “adhesive force” instead of “cohesive force”. “Cohesive” would be only within the matrix.

AR 42:

Cohesive has been  changed to adhesive

Comment 43:

Figure 12         What means “CPS” on the y-axis?

AR 43:

EDAX’s Counts Per Second (CPS) Mapping Smart Feature

Comment 44:

259      What do you mean with “metallic constituents”?

AR 44:

The trace elements present as impurities in the filler

Comment 45:

Fig. 8 - 11       It would be helpful for the reader if more explanation would be given, what can be  seen on the photos (indication with arrows).

AR 45:

The area encircled with the blue lines indicates a good interfacial adhesion between the holding matrix and the fillers. The shiny region marked the blue lines depict the reflection of the light source because of the presence of Diamond-Like Carbon (DLC) entities such as MWCNTs nanoparticles.

Comment 46:

273      61%

AR 46:

Corrected as per the suggestion

Comment 47:

286      What are the other elements: only oxygen is mentioned with a certain concentration. All the other elements are so low in concentration that the effect on the mechanical      properties have been neglected (as stated in line 265.

            Why should a certain content of oxygen have negative impact?

            In the work here only one type of CNT was used. So no results comparing        different  CNT with different composition are given.

AR 47:

The other elements traces are so weak they could not be read even by the smart mapping feature of EDX so the influence of those elements in altering the mechanical properties are neglected.

Comment 48:

What is now the real outcome of the study? Can the results be implemented in

            praxis? Has this been done so far already? What is new with these results compared to what was already known before?

AR 48:

This work determines the optimum reinforcement of nanoparticles required to enhance the strength of nanocomposites. The reinforcement also increases the ductility of the plastic which is very much deserved for polymers in case of structural applications.

We found your comments extremely helpful in enhancing the quality of the paper. The authors are grateful for the time and energy you expended on our behalf.

Reviewer 3 Report

This manuscript is probably more fitting to be submitted to MDPI materials or MDPI polymers rather than MDPI molecules. I rejected the paper mainly for this reason. If the author and editor believe otherwise, I'm happy to read it again to see if there are other issues.

Re ML: The title of "machine learning" is misleading when what you are doing is a polynomial fit of 5 data points? Why is ML even appropriate for this work?? The use of machine learning checks all the boxes for a black-box application, I would suggest you remove the claim of "machine-learning" as it only downgrades the soundness of the paper. 

Author Response

Reviewer #3

The authors take an opportunity to thank the reviewer for the generous words about our manuscript. In the following sections, you will find our responses to each of your comments and suggestions. We found the review comments very useful as we approached our revision, and the authors are grateful for the time and energy you expended on our behalf.

Comment 1: This manuscript is probably more fitting to be submitted to MDPI materials or MDPI polymers rather than MDPI molecules. I rejected the paper mainly for this reason. If the author and editor believe otherwise, I'm happy to read it again to see if there are other issues.

Response: We agree with your comment but this work falls under the aims and scope of this journal headed with nanoscience and material science.

Comment 2: Re ML: The title of "machine learning" is misleading when what you are doing is a polynomial fit of 5 data points? Why is ML even appropriate for this work?? The use of machine learning checks all the boxes for a black-box application, I would suggest you remove the claim of "machine-learning" as it only downgrades the soundness of the paper

Response: Thank you very much for the recommendation, we have implemented your suggestions by replacing the word machine learning with statistical analysis.

We found your comments extremely helpful in enhancing the quality of the paper. The authors are grateful for the time and energy you expended on our behalf.

Round 2

Reviewer 1 Report

The manuscript has been sufficiently improved to warrant publication in Molecules.

Reviewer 3 Report

Happy with the changes and agree with changes made for other reviewers. Best of luck with publication!